# Considerations for Radio Frequency Fingerprinting across Multiple Frequency Channels

**DOI:** 10.3390/s22062111

**Published:** 2022-03-09

**Authors:** Jose A. Gutierrez del Arroyo, Brett J. Borghetti, Michael A. Temple

**Affiliations:** Department of Electrical and Computer Engineering, Air Force Institute of Technology, Wright-Patterson AFB, OH 45433, USA; brett.borghetti@afit.edu (B.J.B.); michael.temple@afit.edu (M.A.T.)

**Keywords:** RF machine learning, deep learning, RF fingerprinting, RFF, specific emitter identification, wireless security

## Abstract

Radio Frequency Fingerprinting (RFF) is often proposed as an authentication mechanism for wireless device security, but application of existing techniques in multi-channel scenarios is limited because prior models were created and evaluated using bursts from a single frequency channel without considering the effects of multi-channel operation. Our research evaluated the multi-channel performance of four single-channel models with increasing complexity, to include a simple discriminant analysis model and three neural networks. Performance characterization using the multi-class Matthews Correlation Coefficient (MCC) revealed that using frequency channels other than those used to train the models can lead to a deterioration in performance from MCC > 0.9 (excellent) down to MCC < 0.05 (random guess), indicating that single-channel models may not maintain performance across all channels used by the transmitter in realistic operation. We proposed a training data selection technique to create multi-channel models which outperform single-channel models, improving the cross-channel average MCC from 0.657 to 0.957 and achieving frequency channel-agnostic performance. When evaluated in the presence of noise, multi-channel discriminant analysis models showed reduced performance, but multi-channel neural networks maintained or surpassed single-channel neural network model performance, indicating additional robustness of multi-channel neural networks in the presence of noise.

## 1. Introduction

Physical-layer emitter identification, known as RFF or Specific Emitter Identification (SEI), is often proposed as a means to bolster communications security [1]. The underlying theory is that the manufacturing processes used for chip components create hardware imperfections that make each emitter unique, irrespective of brand, model, or serial number. These hardware imperfections are akin to human biometrics (e.g., fingerprints) in that they are distinctive and measureable. Imperfections cause small distortions to the emissions of idealized signals, and those signal distortions can be learned by Machine Learning (ML) models to identify emitters solely from their emissions. This is particularly useful for communications security applications where the reported bit-level identity (e.g., MAC Address, Serial Number) of a device cannot or should not be implicitly trusted. Here, RFF can serve as a secondary out-of-band method for identity verification.

Prior related RFF research has trained ML models to identify devices by using bursts received on a single frequency channel. However, modern communications protocols often employ multiple frequency channels to enable simultaneous users and interference avoidance. For example, WiFi (IEEE 802.11 b/g/n) [2] subdivides the 2.4 GHz ISM band into 11 × 20 MHz overlapping channels, ZigBee [3] and Wireless Highway Addressable Remote Transducer (WirelessHART) [4] (i.e., IEEE 802.15.4-based protocols) use the same frequency band but divide it into 15 × 5 MHz non-overlapping channels [5], and Bluetooth [6] uses an even more granular division of 80 × 1 MHz non-overlapping channels.

When the channel changes, the carrier frequency used by both the transmitter and receiver shifts to the center of the new channel. This change in carrier frequency affects the signal distortions because radio hardware components such as Phase-Locked Loops (PLLs), amplifiers, and antennas operate irregularly at different frequencies. Furthermore, the Radio Frequency (RF) environment also varies with frequency channel because different sources of interference are present at different frequencies. Since most RFF research predominantly considers bursts received on a single channel, it is not clear whether the performance achieved by those research efforts generally extends to multiple frequency channel operation.

For instance, when researchers in [7,8,9] collected ZigBee signals, they configured their receivers to capture a narrow bandwidth centered on a single carrier frequency. They used those collections to train RFF models and tested their models on sequestered data from the same collections. No evidence was provided showing that the authors verified model performance across all channels.

In works providing single channel demonstrations, channel carrier frequency details are often omitted, given that the RF signals are commonly down-converted to accommodate baseband fingerprint generation. For example, ZigBee research in [10,11,12,13], and similarly, WirelessHART research in [14], cited the collection bandwidth but omitted carrier frequency information. Furthermore, from an experimental perspective, it is more time-consuming to collect signals across multiple carrier frequencies, given the narrow RF bandwidth limitations of commonly accessible Software-Defined Radios (SDRs). Therefore, the omission of carrier frequency information suggests that researchers in [10,11,12,13,14] did not consider the effects of multi-channel operation.

Finally, there are researchers who implicitly use collections from multiple carrier frequencies but make no explicit declaration in their work. For instance, any researchers using the Defense Advanced Research Projects Agency (DARPA) Radio Frequency Machine Learning (RFML) WiFi dataset, such as [15,16,17], have the ability to account for multiple frequency channels because collections for that dataset were performed with a wide bandwidth. It is not clear whether [15,16,17] deliberately considered frequency channel when selecting training and evaluation datasets.

To our knowledge, no prior work has considered the sensitivity of RFF performance to different frequency channels; our results suggest that frequency channel must be considered. Signal bursts were collected using a wideband SDR receiver, which captured signals from eight IEEE 802.15.4-based devices communicating across 15 frequency channels. Each individual burst was filtered and categorized based on the frequency channel within which it was received. RFF models were trained and tested using data from different channel combinations to evaluate the effects of frequency channel to performance, which was reported using the multi-class MCC. Machine learning models included a Multiple Discriminant Analysis/Maximum Likelihood (MDA/ML) model with expert-designed features, a shallow fully-connected Artificial Neural Network (ANN), a Low-Capacity Convolutional Neural Network (LCCNN), and a High-Capacity Convolutional Neural Network (HCCNN).

The key contributions of our work include:A first-of-its-kind evaluation of the sensitivity of single-channel models to multi-channel datasets. The evaluation suggests that failing to account for frequency channel during training can lead to a deterioration in performance from MCC > 0.9 (excellent) down to MCC < 0.05 (random guess), indicating that single-channel model performance from previous RFF research should not be expected to extend to the multi-channel case (Experiment A).A training data selection technique to construct multi-channel models that can outperform single-channel models, with average cross-channel MCC improving from 0.657 to 0.957. The findings indicate that frequency-agnostic variability can be learned from a small subset of channels and can be leveraged to improve the generalizability of RFF models across all channels (Experiment B).An assessment of multi-channel models against Additive White Gaussian Noise (AWGN) that demonstrated the advantage of multi-channel models in noise performance depended on model type and noise level. Multi-channel neural networks approximately maintained or surpassed single-channel performance, but multi-channel MDA/ML models were consistently outperformed by their single-channel counterparts (Experiment C).

The rest of this paper is structured as follows: an overview of the state-of-the-art in RFF is provided in Section 1.1, and assumptions and limitations of this research are covered in Section 1.2. Our wideband data collection technique is described in Section 2, including how the data were processed for RFF model training. Methodology and results for the three experiments are detailed in Section 3, and study conclusions and potential future work are presented in Section 4.

### 1.1. Related Work

RFF is fundamentally an ML classification problem, where discriminative features are leveraged to distinguish individual devices. Generation and down-selection of the best features remains an open area of research [18]. For instance, researchers have proposed an energy criterion-based technique [19] and a transient duration-based technique [20] to detect and extract features from the transient region of the signal during power-on. Researchers in [14] extract features from the signal preamble region, focusing on down-selecting statistical features to reduce computational overhead while maintaining classification accuracy. Yet another technique presented in [21] extracts features from a 2-dimensional representation of the time series data. Commonly, researchers avoid feature selection altogether by ingesting time series data directly into Convolutional Neural Networks (CNNs), which can be trained to learn the best feature set to be used for the ML task. CNNs have been recently used to classify a large number of devices across a wide swath of operational conditions [17], to verify claimed identity against a small pool of known devices [9], and to measure identifiable levels of I-Q imbalance deliberately injected by the transmitter [15].

Another area of research focuses on bolstering the practicality of deploying RFF mechanisms in operational environments. Notably, ref. [22] explores how neural networks might be pruned to reduce their size and complexity, enabling their deployment to resource-constrained edge devices. Researchers in [23] tackle the need to retrain RFF models whenever a new device is added to the network by employing Siamese Networks, effectively aiding model scalability through one-shot learning. Our work contributes to this effort by comparing performance across four model types with increased levels of complexity, including an expert-feature-based MDA/ML model, a shallow ANN, a low-capacity CNN, and a high-capacity CNN.

Finally, the sensitivity to deployment variability is considered extensively in recent works. For instance, variability stemming from time, location, and receiver configuration are studied extensively in [24], and variability stemming from the RF environment is evaluated in depth by [25,26]. Often, the goal is to remove or reduce environmental effects to bolster classification accuracy, which can be done through data augmentation [27], transmitter-side pre-filtering [28], deliberate injection of I-Q imbalance at the transmitter [15], and through receiver-side channel equalization [17].

Our work extends the research in deployment variability by exploring the impact to model performance from the use of different frequency channels. Consistent with the works by [24,25], we concluded that evaluating models under conditions that are different from training conditions leads to negative impacts to RFF performance. Like [27], we found that adding more variability in our training made the models more generalizable, even to conditions not seen during training.

### 1.2. Assumptions and Limitations

RFF and SEI are broad areas of research that include everything from the fingerprinting of personal and industrial communications devices, to radar and satellite identification. Our work leverages communications devices, but it is likely that any transmitter which operates across multiple carrier frequencies would exhibit the effects highlighted in this research.

The protocol used in this study was WirelessHART, which implements the PHY-layer in the IEEE 802.15.4 specification. That specification divides the 2.4-GHz Industrial, Scientific and Medical (ISM) band into 15 × 5 MHz channels, each with a different carrier frequency. It is not clear whether the performance improvements of the multi-channel models shown in this research depend on the bandwidth of the frequency channel. For instance, models for Bluetooth, which employs narrower 80 × 1 MHz channels, may need more frequency channels or further-spaced channels in the training set to achieve the same levels of performance improvements. The impact of channel bandwidth to multi-channel models is left as future work.

Another key WirelessHART feature is that it allows the use of mesh networking, whereby each device can act as a relay of data for neighboring devices. Mesh networking is also becoming increasingly popular in home automation, particularly with the adoption of new Internet of Things (IoT)-centric protocols such as Thread [29]. The distributed nature of those networks poses a challenge in RFF because there is no centralized endpoint with which all other devices communicate, so there is no ideal centralized location to place the RFF receiver. Our work is limited to the centralized configuration, which assumes that all WirelessHART devices communicate directly with the gateway. Configurations to handle mesh-networking, which could include multi-receiver systems or edge-based RFF, should be explored in future work.

Although our work is limited to preamble-based fingerprinting, in part due to its recent success in classifying WirelessHART devices [13], another highly researched technique is transient-based fingerprinting. Transient-based fingerprinting employs features related to how a device becomes active in preparation for transmission, which has proven fruitful for the purpose of device identification [19,20]. It is likely that transient-based detection will also be affected by carrier frequency, given that the same radio components persist in the transmit chain. Regardless, the study of the effects of frequency channel to transient-based fingerprinting is an interesting area of future work.

In the wideband collection for this work, only one WirelessHART device was configured to communicate at a time. Under real operational conditions, multiple devices would be able to communicate simultaneously on separate frequency channels, introducing the potential for Adjacent Channel Interference (ACI). ACI occurs when energy emitted on one frequency channel leaks into adjacent frequency channels. At a minimum, this energy leakage could raise the noise floor, reducing the Signal-to-Noise Ratio (SNR) and potentially degrading model performance. A study of the specific effects of ACI to model performance is left as future work.

In the end, our goal was to demonstrate that frequency channel can have a significant effect on RFF models in the hopes of encouraging future researchers to take it into consideration.

## 2. Data Collection

A multi-channel WirelessHART dataset was collected and validated for the purpose of this study. Precautions were taken to minimize the effects stemming from the RF environment and receiver, as our focus was the study of transmitter effects. This Section covers the methodology used for collection and pre-processing and includes a description of the training, validation, and evaluation dataset(s).

### 2.1. WirelessHART Communications Protocol

The WirelessHART communications protocol, used by industrial sensors to transmit stateful information (e.g., temperature, humidity, voltage, etc.) between industrial sensors and to human-machine interfaces, implements the Offset-Quadrature Phase Shift Keying (O-QPSK) physical layer from IEEE 802.15.4, the standard for low-rate personal area networks [5]. This is the same standard employed by ZigBee, another common IoT protocol; many RF chips designed for WirelessHART are also compatible with ZigBee applications. WirelessHART uses the first 15 channels defined by the standard on the 2.4 GHz ISM band and employs a pseudo-random frequency hopping scheme, where subsequent transmissions are sent on different frequency channels. Although the channel numbering given by IEEE 802.15.4 ranges from 11 through 25, we arbitrarily number our channels 0 through 14. Then, for a given channel i∈[0,14], its center frequency is $f_{c}\left(i\right)=2405+i\times 5$ MHz, and its bandwidth is $\left(f_{c}-2.5,f_{c}+2.5\right)$ MHz.

WirelessHART was selected as the candidate protocol for collection for several reasons. First, the IEEE 802.15.4-based protocol is representative of many of the basic low-power IoT devices being deployed across the globe at an exponentially increasing rate. Second, it operates in a common frequency band using a manageable number of channels. And finally, recent research in [14] employed MDA/ML using the same WirelessHART devices used in our research, providing a baseline for performance comparison.

### 2.2. Collection Technique

Figure 1 envisions how an RFF-based authentication mechanism might be deployed for WirelessHART. In this configuration, a centrally-located wideband SDR passively captures bursts sent between the WirelessHART sensors and the gateway. Those bursts are offloaded to a monitoring application, which performs RFF and validates the claimed identity of the communicating device. Once the burst is validated, the information contained within it is considered trusted. Our collection setup is based on this conceptual configuration.

WirelessHART devices were allowed to communicate directly with the WirelessHART gateway one at a time, while an SDR captured device emissions. During collection, the device was placed 8 ft. from the gateway, and the SDR antenna was positioned 18 in. from the device. The eight devices observed are listed in Table 1 and included four Siemens AW210 [30] and four Pepperl+Fuchs Bullet [31] devices.

Bursts were captured using a USRP X310 with a 100 MHz bandwidth centered at 2.440 GHz, enabling simultaneous collection of all 15 WirelessHART channels. Burst detection for the wideband data was performed “on-the-fly” by thresholding the power of the received signal. This enabled the collector to be efficient with its limited hard disk space. In particular, given a received signal, $c\left[n\right]=c_{I}\left[n\right]+jc_{Q}\left[n\right]$, the instantaneous power was calculated as ${\left|c\left[n\right]\right|}^{2}={\left(c_{I}\left[n\right]\right)}^{2}+{\left(c_{Q}\left[n\right]\right)}^{2}$. A 100-sample moving average was then used to detect the start and end of each burst using empirically-set thresholds. Buffers of 10 K complex I-Q samples were added before the start and after the end of the burst to aid in SNR approximation, and each detected burst was saved to a new file.

Three precautions were taken in an attempt to minimize effects from the RF environment. First, collections were done within a ranch-style suburban household, far away from other wireless emitters relative to the distance between emitter and receiver. Second, all collections were performed in the same physical location, meaning that any RF effects due to interference with nearby non-emitters would likely manifest in the same way for all devices. Finally, collections were performed in sets of 10 K bursts, started at random times during the day throughout the course of two weeks. This ensured any potential time-dependent sources of interference were well distributed across devices.

Collecting data in sets of 10 K bursts also forced the WirelessHART gateway to assign a new frequency hopping scheme to each device when it re-established communication with the gateway (per the WirelessHART specification [4]). This resulted in a relatively even distribution of bursts across the 15 frequency channels.

### 2.3. Burst Validation

Our approach for burst detection did not guarantee that the received signal came from the expected WirelessHART device; it could be the case that strong interference triggered the burst detector. Further protocol-specific analysis was performed to ensure the validity of each collected burst. The most straightforward way to do this was to detect and verify the structure of the preamble, and subsequently read message-level bits to verify that the transmitted source address corresponded to that listed in Table 1. The process of burst validation consisted of Frequency Correction, Low-Pass Filtering, WirelessHART Preamble Detection and Verification, Phase Correction, and Message Parsing and Address Verification.

#### 2.3.1. Frequency Correction and Low Pass Filtering

Unlike [17], which leveraged the Center Frequency Offset (CFO) as a discriminable classification feature, we chose to remove the CFO because it depends on the characteristics of the receiver, and we are most interested in understanding the impact of carrier frequency to the transmitter. First, bursts were downconverted to baseband using a energy-based coarse estimation of frequency channel. Then, the algorithm presented in [32] was used to quickly approximate the remaining frequency offset between transmitter and receiver by squaring the detected signal and taking a Fast Fourier Transform (FFT). The squaring created peaks at two different frequencies, and taking the average between the peaks gave an estimate of the CFO. Figure 2 shows how two peaks are present after squaring the burst. To correct the frequency offset, the bursts were shifted in frequency such that the two peaks were centered about 0 MHz.

All frequency correction was achieved through multiplication by complex sinusoid. In particular, for a given burst c[n] with estimated center frequency $f_{e}$, the frequency-corrected burst was
(1)$$\begin{array}{c} c_{cor}\left[n\right]=c\left[n\right]e^{-j2\pi (2440\times 10^{6}-f_{e})n} \end{array}$$

After frequency correction, a 2-MHz 4th order Butterworth low pass filter was applied to each burst for noise suppression outside of baseband.

#### 2.3.2. WirelessHART Preamble Detection and Verification

Preamble detection was done in the time domain by correlating the peak-normalized filtered burst with a generated preamble. The index with maximum-amplitude correlation was assumed to be the starting index of the burst preamble. As a measure of similarity, the corresponding correlation coefficient was also used for WirelessHART preamble verification; if the coefficient was too low, the burst was discarded.

#### 2.3.3. Phase Correction

The angle of the correlation coefficient was also an estimate of the phase offset between the burst preamble and the generated preamble. It was used to correct phase through multiplication by complex exponential. Given a burst c[n] with phase offset θ, the phase-corrected burst was
(2)$$\begin{array}{c} c_{cor}\left[n\right]=c\left[n\right]e^{-j\theta } \end{array}$$

A note on frequency and phase correction: in operational receivers, both phase and frequency correction are done via PLLs [33]. PLLs continuously tune the receiver center frequency to drive the phase offset between transmitter and receiver to zero. This is undesirable for the RFF application because the distortions in phase and frequency, which are suppressed by the PLL, could be leveraged for device discrimination.

#### 2.3.4. Message Parsing and Address Verification

The downconverted, phase-corrected burst was demodulated, and the symbols were mapped to message bits. The source address was extracted from the parsed message and compared to the known device addresses listed in Table 1. If the address matched the expected device address, the burst was considered valid.

### 2.4. SNR Estimation

SNR estimation of the validated bursts was necessary to enable model evaluation against noise. All SNR estimates were performed after the 2-MHz low pass filter was applied.

Noise power and signal-plus-noise power were estimated on a per-burst basis directly from the captured burst. The signal-plus-noise region from which signal-plus-noise power ($P_{S+N}$) was estimated was defined as the 1605 samples that made up the preamble. The preceding 160 samples were deemed a buffer region, where none of the samples were used for any power estimation. The noise region for estimating noise power ($P_{N}$) was defined as the 1605 samples immediately preceding the buffer region. Figure 3 shows these three regions for an example burst. For a given complex burst, $c\left[n\right]=c_{I}\left[n\right]+jc_{Q}\left[n\right]$, the estimated powers and SNR (in dB) are
(3)$$\begin{array}{cc} P_{N} & =\frac{\sum_{n\in \mathrm{noise}\;\mathrm{rgn}}\left(c_{I}{\left[n\right]}^{2}+c_{Q}{\left[n\right]}^{2}\right)}{1605} \end{array}$$
(4)$$\begin{array}{cc} P_{S+N} & =\frac{\sum_{n\in \mathrm{sig}+\mathrm{noise}\;\mathrm{rgn}}\left(c_{I}{\left[n\right]}^{2}+c_{Q}{\left[n\right]}^{2}\right)}{1605} \end{array}$$
(5)$$\begin{array}{cc} SNR & =10\log_{10}\left(\frac{P_{S}}{P_{N}}\right)=10\log_{10}\left(\frac{P_{S+N}-P_{N}}{P_{N}}\right). \end{array}$$

Figure 4 shows the per-class range and average SNRs estimated on the full set of collected bursts. The average burst SNR across all devices and all channels was 41.4 dB.

### 2.5. Datasets

Once all bursts were validated, datasets were created for the purpose of model training, validation, and evaluation. The large pool of valid bursts were randomly sampled to create datasets that were balanced with respect to the classes and to the frequency channels. Table 2 outlines the fundamental datasets for our work. The experiments leveraged subsets of these datasets (e.g., by training with data from only one channel), but the delineation between the sequestered training, validation, and evaluation datasets remained.

## 3. Experiments

Three experiments were designed to (i) assess the sensitivity of single-channel models to a multi-channel dataset, (ii) determine whether carefully-constructed multi-channel models generalized better than single-channel models, and (iii) determine whether multi-channels models gained more resiliency to noise. This section provides a description of the four models used across all tests, followed by individual methodologies, results, and analyses for the three experiments.

### 3.1. RFF Models

Four typical RFF models with varying levels of complexity were selected as candidates for experimentation in this work. All four model types were used in all three experiments. This section describes the structure of each model in detail.

Multiple Discriminant Analysis/Maximum Likelihood. This model type leverages the commonly-used Time-Domain Distinct Native Attribute (TD-DNA) feature set [10,34,35], as depicted in Figure 5, which was recently used by Rondeau et al. on single-channel WirelessHART bursts [14]. TD-DNA features are statistics calculated for a set of signal subregions (defined by $N_{R}$) after a number of signal transforms. The features are dimensionally reduced through Fisher Transform and fed in to a standard Quadratic Discriminant Analysis (QDA) model for classification. Including dimensionality reduction and discriminant analysis, the total number of trainable parameters is 1757.Fully-Connected Artificial Neural Network. Figure 6 shows the ANN, which operated directly on raw complex I-Q burst data input to the network on two independent data paths (one for I and one for Q). The two-dimensional input was flattened into a single 3210-wide vector which was fed to the first fully-connected layer. Even with 207,848 trainable parameters (10 times that of MDA/ML), this particular ANN was a shallow, low-complexity alternative to the other two neural networks.Low-Capacity Convolutional Neural Network. Like the ANN, the LCCNN presented in Figure 7 operated directly on raw complex I-Q burst data. The network comprised eight Conv1D layers that applied a total of 112 digital filters and two fully-connected layers that mapped the filter output to a four-dimensional latent space. The Softmax output layer was used for classification, where class prediction was determined by the node with largest output value. Because of its convolutional layers, the LCCNN is able to add depth to the ANN with 232,156 trainable parameters.High-Capacity Convolutional Neural Network. This high-capacity model was inspired by ORACLE, a CNN that extracts and classifies injected I-Q imbalance [15]. This model carried significantly more capacity, with 14 hidden layers comprising four pairs of 128-filter Conv1D layers that applied a total of 1024 digital filters, and two fully-connected layers that mapped the filter outputs to a 64-dimensional latent space. Like in the LCCNN, the input layer contained two independent input paths, and the output Softmax layer contained one node for each of the eight classes in this study. This is the most complex model, requiring 3,985,544 trainable parameters, approximately 20 times as many as the ANN and LCCNN. Figure 8 depicts the HCCNN.

MDA/ML models were trained by fitting a QDA model onto the dimensionally-reduced TD-DNA features. All neural networks were trained using Stochastic Gradient Descent with the Adam optimizer and learning rate of $1\times 10^{-4}$, with batch size of 512 observations. An early stopping callback was used to monitor the performance on the validation set, where training was stopped after 10 epochs of no improvements. The model weights with best validation set performance were restored and this was used as the final trained model.

**Figure 5 sensors-22-02111-f005:**
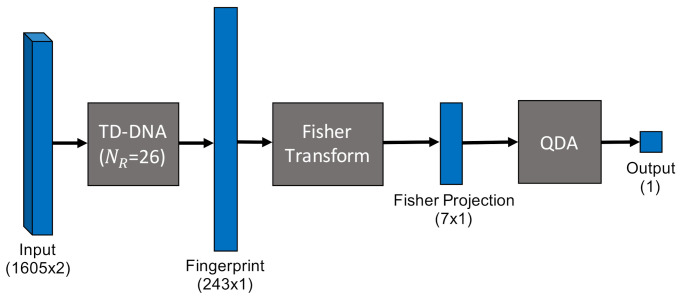
MDA/ML QDA model with TD-DNA feature set with 26 subregions.

**Figure 6 sensors-22-02111-f006:**
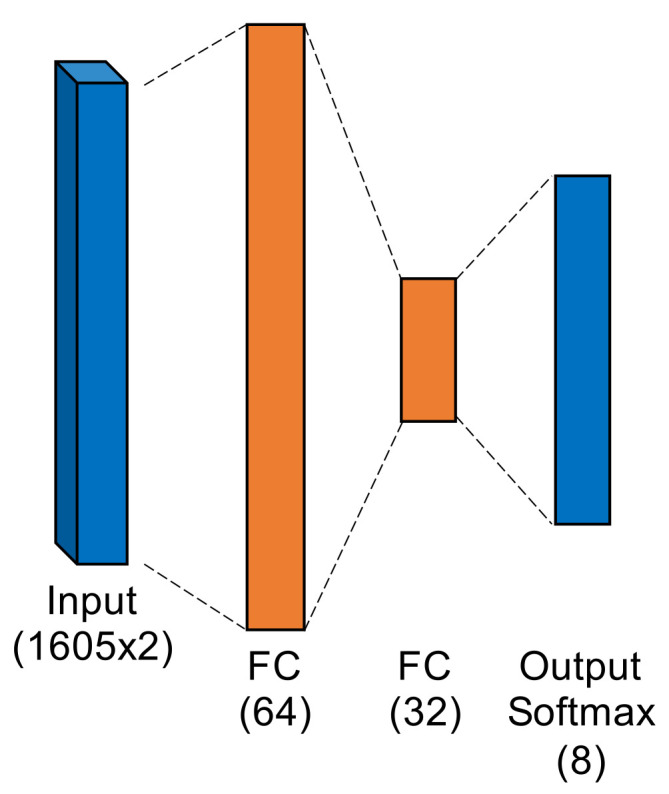
Fully-Connected Artificial Neural Network.

**Figure 7 sensors-22-02111-f007:**
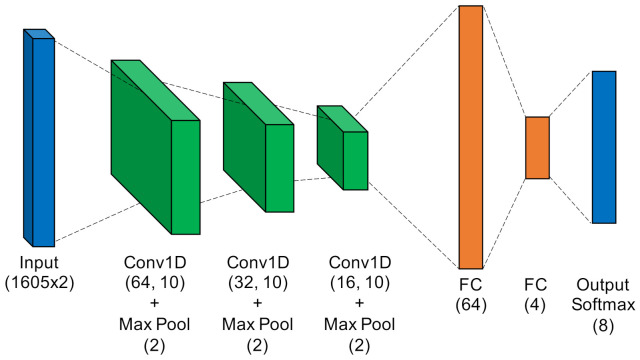
Low-Capacity Convolutional Neural Network.

**Figure 8 sensors-22-02111-f008:**
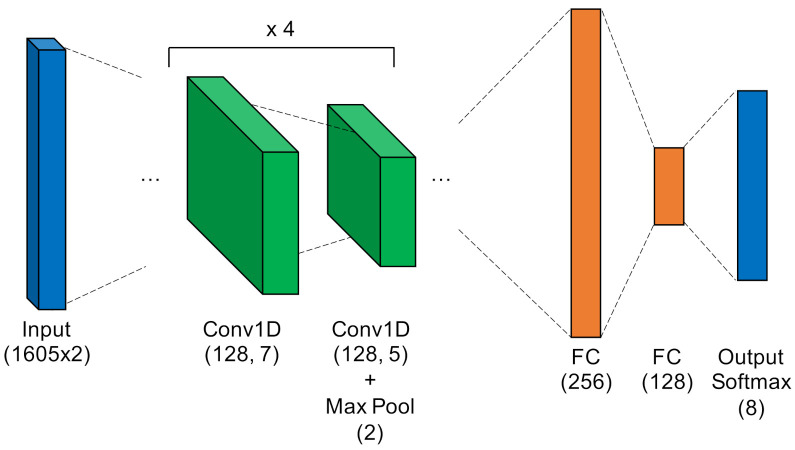
High-Capacity Convolutional Neural Network.

### 3.2. Performance Metric: Matthews Correlation Coefficient (MCC)

Performance for RFF models is typically reported as per-class classification accuracy, which is the accuracy averaged across all classes. For *K* classes, per-class classification accuracy degrades to 1/K when models perform no better than random guess. It is therefore difficult to grasp model performance without first knowing how many classes were used to train the model. One way to address this problem is to standardize performance by the number of classes through the use of MCC.

MCC is a performance metric first designed for binary class classification models, derived by Matthews as a discrete version of the correlation coefficient [36]. An MCC value of 1.0 indicates perfect correct model performance, 0.0 indicates performance no better than random guess, and −1.0 indicates perfect incorrect model performance. The multi-class case, derived by Gorodkin [37], preserves the −1.0 to 1.0 quantitative performance characteristics. For *K* classes, MCC is calculated as
(6)$$\begin{array}{c} MCC=\frac{cs-\sum_{k}p_{k}t_{k}}{\sqrt{s^{2}-\sum_{k}{\left(p_{k}\right)}^{2}}\;\sqrt{s^{2}-\sum_{k}{\left(t_{k}\right)}^{2}}} \end{array}$$
where $t_{k}$ is the number of occurrences for class k∈[0,1,2,…,K−1], $p_{k}$ is the number of times class *k* was predicted, *c* is the total number of correct predictions, and *s* is the total number of predictions [38]. For our experiments, we use K=8 classes and report all model performance via MCC.

### 3.3. Experiment A: Single-Channel Models

The goal in this experiment was to determine whether the performance of models trained on a single channel (i.e., “single-channel models”) extends to other frequency channels. To that end, single-channel models were evaluated against a dataset containing bursts from multiple channels, and the per-channel evaluation performance was reported.

#### 3.3.1. Methodology

The training and validation datasets described in Section 2.5 were subdivided into 15 subsets, one for each channel, yielding a total of 5 K training observations and 500 validation observations per device in each subset. Then, the four models were trained with each subset to create a total of 60 single-channel RFF models. All models were evaluated on the full evaluation dataset described in Section 2.5, which contained bursts from all 15 channels. MCC was calculated using Equation (Equation 6).

#### 3.3.2. Results and Analysis

Figure 9, Figure 10 and Figure 11 show the performance for single-channel models trained on data from Channel 0, Channel 7 and Channel 14, respectively. Plots for the remaining single-channel models have been omitted for brevity, given these three figures are representative of the overarching observations for this experiment. Performance was reported individually for each channel in the evaluation set to show how well the models operated outside their training scope. Note that performance always peaked at the channel on which the models were trained and deteriorated when the evaluation channel was farther away (in frequency) from the training channel.

As expected, none of the models generalized across all channels when trained on data from a single channel, but some models did generalize better than others. In particular, when the models were trained on data from Channel 7, they generalized well across a wide swath of channels (e.g., Channels 5–10), whereas when the models were trained on Channel 14, performance only roughly generalized to Channel 13 and only for MCA/ML and HCCNN. In the worst case, signals from distant channels were classified by the Channel 14 model with success no better than random guess.

Between the model types, the MDA/ML model and the HCCNN model were the most competitive, with the MDA/ML model performing best most of the time. The LCCNN and ANN performed similarly most of the time, but the LCCNN was especially bad at generalizing when it was trained on Channel 14. These observations suggest that the MDA/ML and HCCNN model were better at inherently targeting variability that existed irrespective of frequency channel.

In general, frequency-dependent signal distortions appeared to be on a spectrum, where signals from nearby channels exhibited similar distortions. However, it is not clear how the channel width might affect those distortions. It could be the case that with narrower channels, like the 1 MHz channels in Bluetooth, performance extends across more channels. Such exploration of the effects of channel width are interesting future work.

Additionally, recall that the transmitter, RF environment, and receiver distortions were not decoupled. Thus, it is possible that a non-trivial part of the variability between frequency channels was imposed by non-transmitter sources. Finding a way to decouple (or at least reduce the coupling) across the three sources would enable more flexible applications (e.g., multi-receiver narrowband systems) and could help researchers more precisely target the signal alterations imposed by hardware components in the transmitter. Regardless of its source, the variability must be considered to achieve good RFF performance.

A natural follow-up experiment was to include data from multiple channels in the training set to determine if this improved model generalizability. This experiment is covered in the following section.

### 3.4. Experiment B: Multi-Channel Models

The focus of Experiment A was to demonstrate that single-channel models did not always perform well across all frequency channels. In Experiment B, the goal was to determine whether including bursts from multiple channels during training improves performance. During training, we deliberately use data from an increasing number of channels, relatively spread throughout the 80 MHz band to create “multi-channel models.” These models were tested against the same evaluation set from Experiment A, which included bursts from all 15 channels.

#### 3.4.1. Methodology

The same four models presented in Section 3.1 were used in Experiment B. Training was done using 11 datasets assembled from portions of the full training set described in Section 2.5, but special consideration was taken with respect to the number of observations in training sets.

Generally, the performance of ML models is influenced by the number of observations provided during training. To enable comparison between multi-channel models from Experiment B and single-channel models from Experiment A, the size of the training datasets was limited to no more than 5000 observations per device (i.e., the size of training sets in Experiment A). Concretely, two-channel data subsets contained 2500 observations per device per channel (5000 observations/device in total), three-channel data subsets contained 1666 observations/device per channel (4998 observations/device), four-channel subsets had 1250 observations per device per channel (5000 observations/device), and the all-channel (i.e., 15-channel) subset contained only 333 observations per device per channel (4995 observations/device). Channel combinations were selected to explore how channel coverage impacted performance. The 11 data subsets used for Experiment B are listed in Table 3. Evaluation was done using the same dataset from Experiment A, which contained bursts from all 15 channels.

#### 3.4.2. Results and Analysis

A representative sample of the performance results for the multi-frequency models are depicted in Figure 12, Figure 13, Figure 14 and Figure 15.

Comparing the Channel 14 single-channel models from Experiment A in Figure 11 with the 2-channel models in Figure 12, it is immediately evident that adding even one more channel to the training set improved generalizability, regardless of model type. With the addition of Channel 0 to the training set, the worst-case performance across all models improved from MCC=−0.23 to MCC=0.716. Note that model performance improved across all channels, even if the channels were not explicitly included in the training set.

Table 4 and Figure 16 summarize the combined performance of models from Experiment A and Experiment B. A single MCC metric was calculated for each trained model by aggregating the per-channel results, and ${\mathrm{MCC}}_{avg}$ was calculated by averaging across all models of the same type that employed the same number of channels. For the 15-channel case, the reported ${\mathrm{MCC}}_{avg}$ is the performance of the sole 15-channel model.

Two trends are evident from these metrics: (i) model performance generally improved when channels were added to the training set, and (ii) the performance of 4-channel models approached the performance of 15-channel models. Note that even the worst performing single-channel model type, i.e., the LCCNN, improved its ${\mathrm{MCC}}_{avg}$ from 0.657 to 0.957 with only three additional channels in the training set. The one exception was with the ANN, for which the 15-channel ${\mathrm{MCC}}_{avg}$ was 0.006 units lower than the 4-channel ${\mathrm{MCC}}_{avg}$. This small gap in performance might be attributed to the variability stemming from the randomized initial model weights before training, or from the fact that the 15-channel “average” included only one model—retraining the 15-channel ANN may yield slightly better results.

Regardless, these trends again support the existence of frequency-irrespective variability. Multi-channel models were better suited to learn that variability, even with limited (i.e., 4-channel) exposure to the spectrum. Furthermore, the MDA/ML model consistently generalized better than its neural network counterparts, suggesting that the frequency-irrespective variability in our particular experimental setup can be effectively extracted through the TD-DNA fingerprint generation process.

In practice, it is desirable for RFF models to perform well across all frequency channels, but that is not the only requirement. Models should also perform well under the presence of environmental noise. Thus, researchers often report model performance across multiple levels of noise, modeled as AWGN. Experiment C explores whether multi-channel models gain any performance advantages over single-channel models under varying noise conditions.

### 3.5. Experiment C: Gains in Noise Performance

The goal of the final experiment was to determine whether multi-channel models gained any performance advantages over single-channel models in noisy RF environments. Tested models included the Channel 7 single-channel models from Experiment A, and new “all-channel/all-data” multi-channel models built exclusively for this experiment. All models were evaluated with bursts from Channel 7 with varying SNR levels adjusted synthetically through the addition of power-scaled AWGN.

#### 3.5.1. Methodology

Two datasets were used for training: a single-channel set, where the models were only exposed to Channel 7 data (5 K observations/device), and a multi-channel set, which included the full training dataset (75 K observations/device). Performance was captured on a subset of the full evaluation set with only Channel 7 bursts (1 K observations/device). The full training dataset was used to maximize the exposure of multi-channel models to Channel 7 data, enabling them to better compete against the single-channel models. Multi-channel models were at an inherent disadvantage because they had to overcome both channel and noise, whereas single-channel models only had to overcome noise.

Each model was trained and evaluated at the same SNR level, which was adjusted to simulate increasingly harsh operational environments. Individual bursts were adjusted to the desired SNR through the addition of AWGN. For each burst, a noise realization was generated from a normal distribution and filtered using the same 2-MHz low pass filter from burst processing. Then, the power of the noise realization was scaled such that when it was added to the burst, the desired SNR was achieved. Models were re-trained at each SNR level, and MCC was calculated after aggregating the results across 100 noise realizations. Performance was reported for SNR∈[5,6,7,8,9,10,12,14,16,18,25,25,30,35,Max], where “Max” means no SNR adjustment was made (i.e., capture SNR).

Note that since noise could only be added (not removed), the per-burst SNR could only ever be decreased. As an example, Figure 17 shows the same bursts from Figure 4 after they were adjusted to a maximum SNR of 35 dB, resulting in a mean SNR of 34.8 dB across all data. In that case, bursts that had SNR lower than 35 dB remained unchanged.

#### 3.5.2. Results and Analysis

Figure 18, which depicts noise performance of the LCCNN models, is representative of the results across all model types. As expected, MCCs for both the single- and multi-channel models worsened as SNR was decreased. To help determine whether multi-channel models gained an advantage against noise, we define a new metric, ${\mathrm{MCC}}_{\Delta }$, as the difference between multi-channel performance and single-channel performance for a given model type, i.e.,
(7)$$\begin{array}{c} {\mathrm{MCC}}_{\Delta }=\mathrm{MCC}\left(\mathrm{multi-channel}\right)-\mathrm{MCC}\left(\mathrm{single-channel}\right), \end{array}$$
for which a positive ${\mathrm{MCC}}_{\Delta }$ implies better multi-channel performance. Figure 19 illustrates ${\mathrm{MCC}}_{\Delta }$ for the four models across varying SNR levels.

The advantage of multi-channel models depended on model type and SNR level. At high SNR levels (SNR > 20 dB), ${\mathrm{MCC}}_{\Delta }$ was generally stable for all model types. In that region, the multi-channel CNNs matched or beat single-channel CNNs, but the single-channel ANNs and MDA/ML models bested their multi-channel counterparts. With mid-level SNRs (10 dB < SNR < 20 dB), ${\mathrm{MCC}}_{\Delta }$ for the three neural networks fluctuated between positive and negative, suggesting no clear advantage for multi-channel models. Notably, single-channel MDA/ML models thrived in this region, surpassing multi-channel models by up to 0.13 units. Finally, at low SNR levels (SNR < 10 dB), multi-channel models for all four model types showed some advantage over single-channel models, though arguably, the performance in this region was already too weak (MCC≲0.5) to be practical for RFF applications.

For the neural networks (i.e., CNNs and ANN), the frequency-irrespective variability learned by the multi-channel models enabled them to approximately maintain or surpass single-channel model performance. Conversely, the single-channel MDA/ML models consistently outperformed their multi-channel counterparts in the presence of noise. It could be that frequency-specific variability, which was deliberately ignored by multi-channel models, allowed some of the single-channel MDA/ML models to overfit the training channel, giving them an advantage against random noise.

## 4. Conclusions and Future Work

Modern communications protocols often employ multiple frequency channels to enable simultaneous user operation and mitigate adverse interference effects. Although recent RFF research targets devices that implement these protocols, the direct applicability of these proof-of-concept works is generally limited given that they train RFF models using bursts received on a single channel. Because the signal distortions leveraged by RFF models are linked to the radio hardware components, and those components operate irregularly across different frequencies, practical RFF models must account for multiple frequency channels. Using WirelessHART signal bursts collected with a wideband SDR, our work demonstrated that RFF model performance depends on the frequency channels used for model training.

Candidate models, including MDA/ML using expert-aided features, a fully-connected ANN and two CNNs, were evaluated across several training-evaluation channel combinations. Performance of single-channel models did not always generalize to all frequency channels. In the most disparate case, one of the single-channel models performed almost perfectly (MCC > 0.9) on its training channel and no better than random guess (MCC < 0.05) on a non-training channel. Often, models performed well on the training channel and relatively well on the adjacent channels, but deteriorated outside of that scope. This suggests that signal distortions were continuous with respect to frequency, i.e., nearby channels exhibit similar distortions.

When data from multiple channels were included in the training set, the multi-channel models generalized better across all channels, achieving adequate performance even when just a small subset of channels were included (i.e., four of the 15). In the worst case, the average MCC for LCCNN models improved from 0.657 in the single-channel configuration to 0.957 in the 4-channel configuration, again implying bolstered performance across all channels. This finding suggests that there existed frequency-irrespective variability that could be learned by the models and used for RFF.

The performance advantage of multi-channel models under noisy conditions depended on model type and SNR level. Multi-channel neural networks (i.e., CNNs and ANN) were able to approximately maintain or surpass single-channel model performance across most SNR levels, but multi-channel MDA/ML models were consistently outperformed by their single-channel counterparts. It could be that the frequency-specific variability available to the single-channel MDA/ML models caused them to overfit the training channel, giving them an advantage against random noise.

One interesting area of future work would be to explore how the bandwidth and spectrum location of the frequency channels used in training affects multi-channel performance. Each additional channel included in the training set exposed the RFF models to an additional 5-MHz “chunk” of that spectrum. This additional exposure enabled models to learn frequency-agnostic variability, making them generalize better across all frequencies. We found that 20 MHz (i.e., four WirelessHART channels) of exposure spread throughout the 80 MHz band was enough to achieve frequency channel-agnostic performance. Other common communications protocols employ channels of different sizes; e.g., Bluetooth channels are 1 MHz wide, and typical WiFi channels are up to 20 MHz wide. It could be the case that more Bluetooth channels and fewer WiFi channels would be needed to achieve generalizable multi-channel model performance because of the difference in channel bandwidth. Further study of the effects of channel bandwidth and spectrum location to frequency-irrespective variability remains an area of future work.

Another area of future work would be to address the radio limitations for practical RFF applications. As discussed, modern IoT protocols enable mesh networking, whereby each endpoint in the network can relay data to and from its neighbors. A practical RFF solution must be able to target all of these data transfers to be useful for security. One solution would be to include RFF capabilities within the individual endpoints, as proposed by researchers in [22]. To that end, our work explored the use of low-complexity models in a multi-channel configuration (e.g., MDA/ML or LCCNN and found them to be generally adequate under most conditions, as long as they were trained using multiple channels. Indeed, this type of deployment is the long-term vision for wireless security, but it does not address the devices that are already deployed and operational.

A stopgap solution would be to deploy more RFF-capable SDRs, forming multi-receiver RFF systems. Multi-receiver systems could also be useful in non-mesh configurations if individual SDRs cannot cover all frequency channels. The key challenge would be to find a way to share RFF models across radios to avoid the tedious collection and training effort that would come with scale. One approach may be to combine bursts collected from multiple receivers, similar to our multi-channel approach, whereby the RFF models could learn receiver-irrespective variability. Notably, this effort would also aid in decoupling signal distortions imposed by the receiver from those imposed by the transmitter and RF environment, further adding to its value as future work.

Finally, with the extension of RFF to multi-channel configurations, the effects of ACI to RFF model performance should be explored. When multiple devices communicate simultaneously on different frequency channels, the potential exists for some of the energy in one channel to leak to adjacent channels. At a minimum, this energy leakage could raise the noise floor, reducing SNR and potentially degrading model performance. Understanding the extent to which ACI can affect model performance will therefore be critical in the deployment of RFF models to real operational environments.

RFF models continue to offer an attractive out-of-band method for wireless device authentication, especially as a component in the defense-in-depth security paradigm. As modern protocols grow in operational complexity, the variability of signal distortions across these expanded modes of operation must be considered to achieve the most effective and generalizable RFF systems.

## Figures and Tables

**Figure 1 sensors-22-02111-f001:**
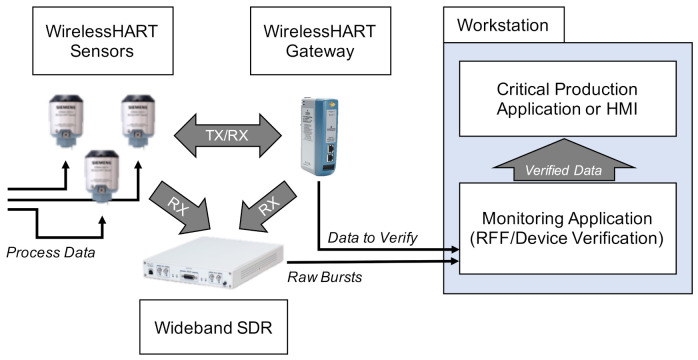
Envisioned use case for RFF-based authentication of WirelessHART devices. Our experimental setup mimicked this configuration.

**Figure 2 sensors-22-02111-f002:**
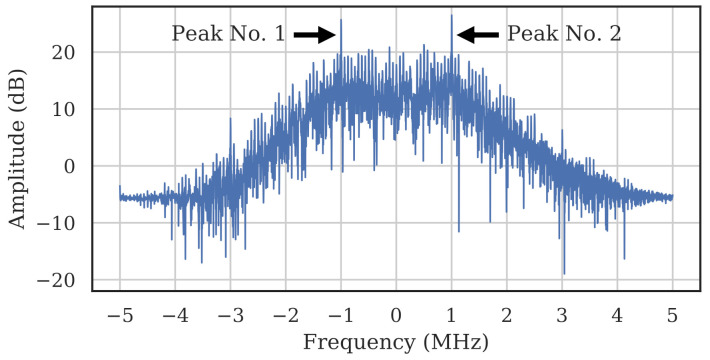
FFT of the square of a WirelessHART burst with yielded peaks. Centering the peaks about 0 MHz center-aligns the carrier and corrects the frequency offset.

**Figure 3 sensors-22-02111-f003:**
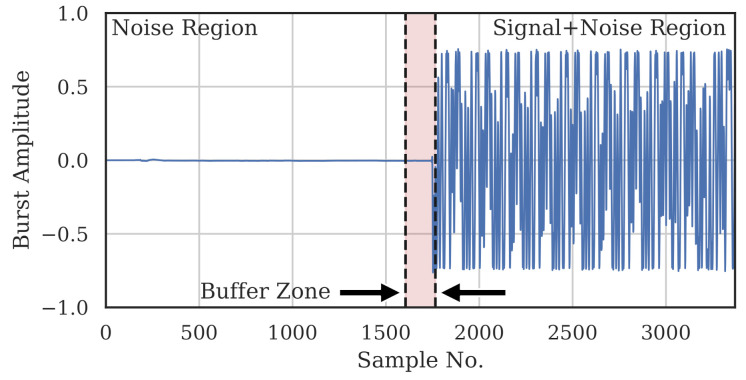
Sample burst depicting the noise region (1605 samples), buffer zone (160 samples) and signal-plus-noise region (1605 samples) used for SNR estimation. For clarity, only the I-component is shown, but the power was calculated using the complex samples.

**Figure 4 sensors-22-02111-f004:**
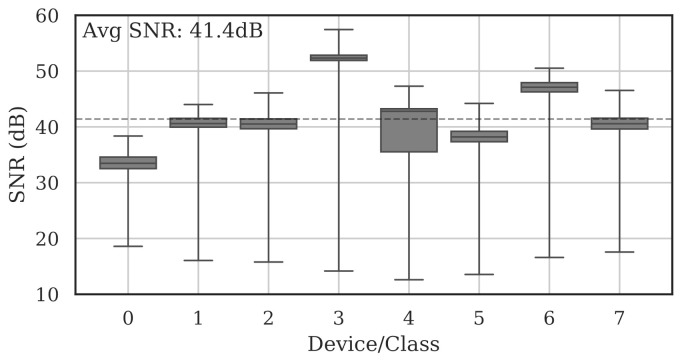
Summary of estimated collection SNRs for all devices across all channels. As denoted by the dotted line, the average SNR was 41.4 dB.

**Figure 9 sensors-22-02111-f009:**
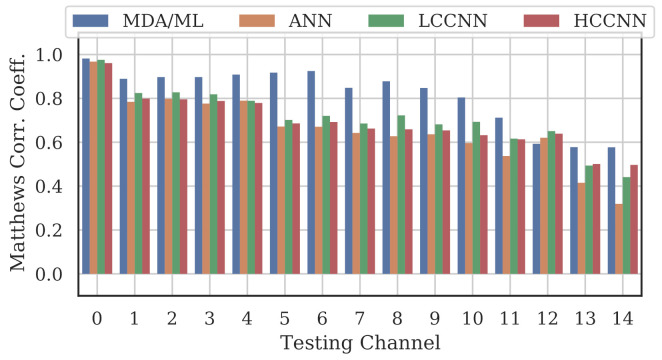
Performance of single-channel models trained with data from Channel 0.

**Figure 10 sensors-22-02111-f010:**
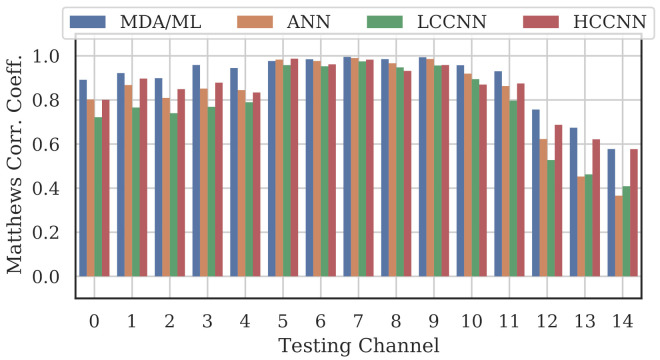
Performance of single-channel models trained with data from Channel 7.

**Figure 11 sensors-22-02111-f011:**
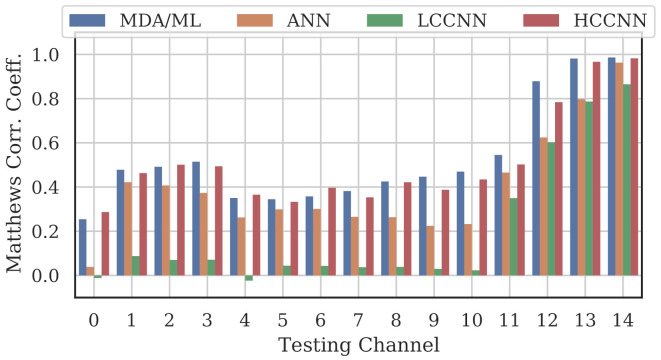
Performance of single-channel models trained with data from Channel 14.

**Figure 12 sensors-22-02111-f012:**
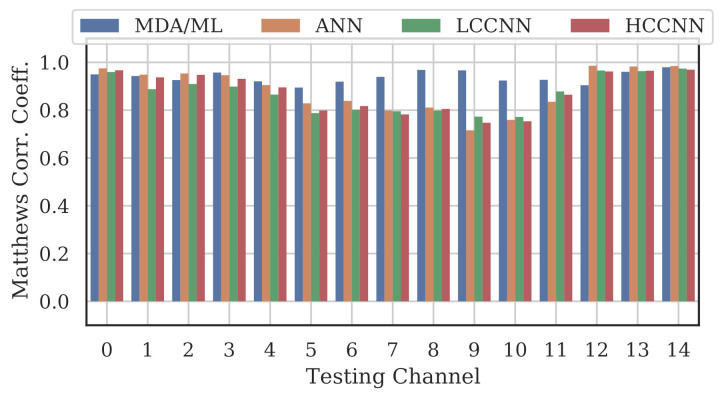
Performance of multi-channel models trained with data from Channels 0 and 14.

**Figure 13 sensors-22-02111-f013:**
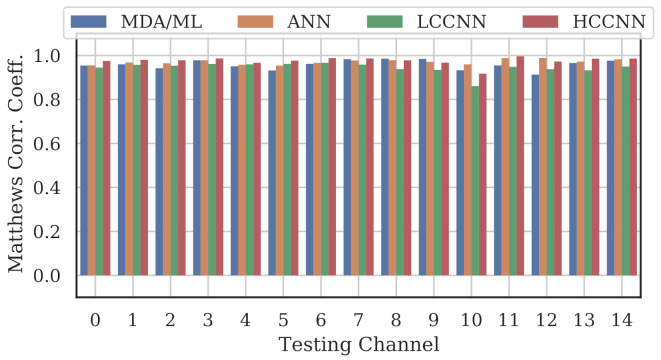
Performance of multi-channel models trained with data from Channels 0, 7 and 14.

**Figure 14 sensors-22-02111-f014:**
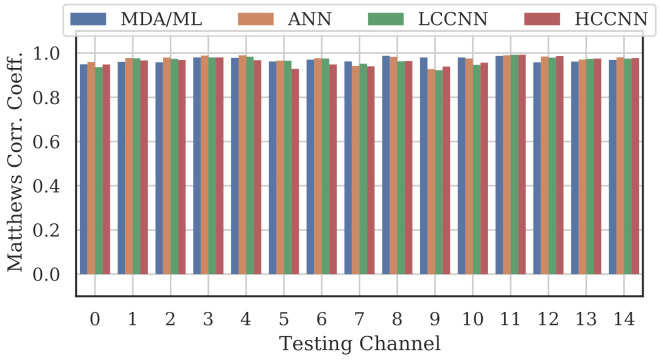
Performance of multi-channel models trained with data from Channels 0, 4, 10 and 14.

**Figure 15 sensors-22-02111-f015:**
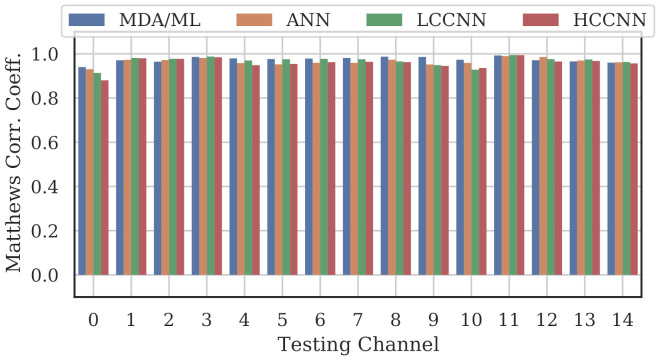
Performance of multi-channel models trained with data from all channels.

**Figure 16 sensors-22-02111-f016:**
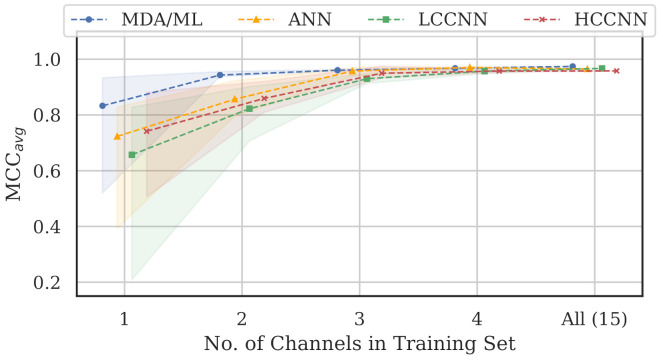
Performance summary for all models as the number of channels in the training was increased. Trend lines represent ${\mathrm{MCC}}_{avg}$, and color bands represent the range of MCC values at that number of training channels. Notably, all-channel performance was achieved with only four channels included in the training set.

**Figure 17 sensors-22-02111-f017:**
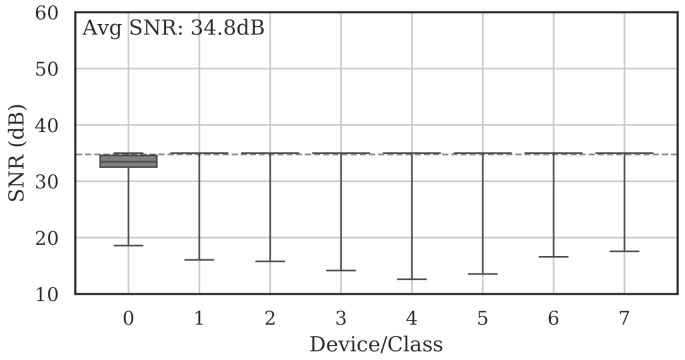
Summary of estimated collection SNRs for all classes across all channels after adjusting SNR to 35 dB. The mean SNR dropped to 34.8 dB, as denoted by the dotted line.

**Figure 18 sensors-22-02111-f018:**
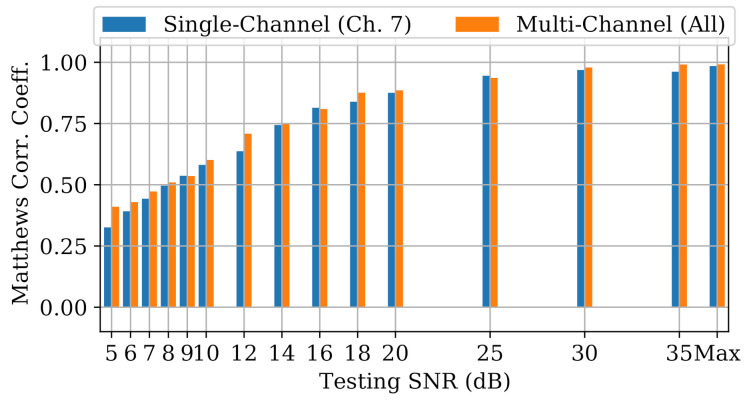
Performance of single-channel and multi-channel LCCNN models at varying SNRs.

**Figure 19 sensors-22-02111-f019:**
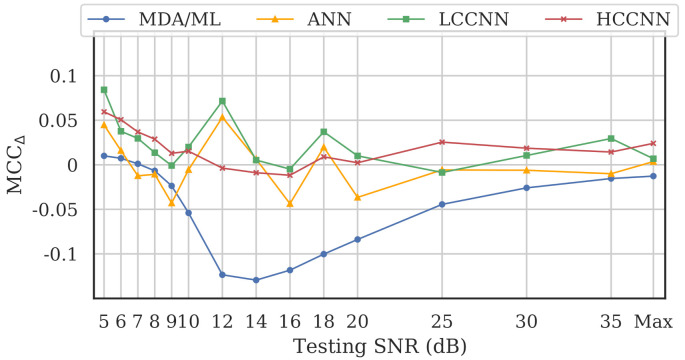
Performance difference between multi-channel models and single-channel models across varying SNR levels, where ${\mathrm{MCC}}_{\Delta }>0$ implies better multi-channel performance. Multi-channel neural networks (i.e., CNNs and ANN) approximately maintained or outperformed their single-channel counterparts, but multi-channel MDA/ML models did not.

**Table 1 sensors-22-02111-t001:** Serial numbers and source addresses for the eight WirelessHART devices.

Device Number	Manufacturer	Serial Number	Hex Source Address (Assigned by Gateway)
0	Siemens	003095	0002
1	Siemens	003159	0005
2	Siemens	003097	0006
3	Siemens	003150	0003
4	Pepperl+Fuchs	1A32DA	0004
5	Pepperl+Fuchs	1A32B3	0007
6	Pepperl+Fuchs	1A3226	0008
7	Pepperl+Fuchs	1A32A4	0009

**Table 2 sensors-22-02111-t002:** Fundamental datasets used in this work, broken down by class and channel.

No. of Observations	Training Set	Validation Set	Evaluation Set
per Device per Channel	5000	500	1000
per Channel	40,000	4000	8000
per Class	75,000	7500	15,000
in Total	600,000	60,000	120,000

**Table 3 sensors-22-02111-t003:** Summary of datasets used for Experiment B.

No. of Chans.	Chans. in Datasets	Observations per Chan. per Device	Observations per Device	Total
2	[0,14], [1,13], [2,12], [3,11]	2500	5000	40,000
3	[0,7,14], [1,7,13], [2,7,12], [3,7,11]	1666	4998	39,984
4	[0, 4, 10, 14], [1, 5, 9, 13]	1250	5000	40,000
All	[0,1,2,…,14]	333	4995	39,960

**Table 4 sensors-22-02111-t004:** Summary of ${\mathrm{MCC}}_{avg}$ for all models from Experiments A and B. In general, MCC improved as the number of channels in the training set were increased.

No. of Channels	${\mathrm{MCC}}_{avg}$			
	MDA/ML	ANN	LCCNN	HCCNN
1	0.833	0.723	0.657	0.742
2	0.943	0.857	0.823	0.859
3	0.961	0.957	0.930	0.950
4	0.967	0.970	0.957	0.958
15 (All)	0.974	0.964	0.967	0.958

## Data Availability

The data presented in this study are available on request from the corresponding author. The data are not publicly available due to the potential leakage of private information. Bursts were captured on the commonly-used 2.4 GHz ISM band at an off-campus location and may inadvertently contain private Bluetooth and/or WiFi communications.

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
