# Peer review of "Considerations for Radio Frequency Fingerprinting across Multiple Frequency Channels"

_sensors, 2022, doi:10.3390/s22062111_

Round 1

Reviewer 1 Report

The paper presents the results of an experimental work toward exploiting how RF fingerprinting peformance varies when single and multiple channel dateset are used in classification, for which four models are impemented. The findings are valuable based on the comparative results.

 I have the following major comments:

First of all, the literature survey would be extended to (well-)position the paper in the RFF domain. I have given several references with my comments below but it is not limited to those papers. 

Next, the records include preamble phase of the signal only. Transients seem to be discarded by employing buffering concept (fig.5). However, it is known that unique features could be extrated from this region, mostly. Knowing that some computationally fast and accurate transient detection techniques are developed [1], what do the authors think about this (SDR seems to have capability to detect transients!)? Can you comment and discuss this, briefly?

Another issue is partial equalization, the carrier de-offsetting [2]. This could be further dicussed with the reference. Moreover, use of single and multi-channel dataset may also be exploited with this reference. For practical implementation point of view, can you comment on implementing RFF in edge stage [3], possiblility with multi channel records?

One another issue; the authors use very short links as they attempt to demonstrate multi-channel recording effects. However, channel impairments would still be applicable [4].  How such multi-channel concept is applicable in longer ranges, and fading channels?

Finally, a recent work would be valuable [5] to discuss narrowband and dense channelization of some other devices.

It seems that records from 3 channels (0,7,14) would mostly work well over 80 MHz ISM band? Is that what we can conclude? It would be higher when narrowband systems are considered (like BT)?

Minor comments:

  • Table I would be converted simply to a form f_c(i)=2405+i*5 where i=0:14.
  • Line:406, "synthesized preamble" may refer to generate synthetic data. but it is not case here?
  • Something wrong with eqn(10-11) c_i and c_q?
  • Table 3 and in other texts, "examples" is not common. Its it "samples"

  1. Mohamed I. et al, On the Perfomance of Energy Criterion Method in Wi-Fi Transient Signal Detection, 2022.
  2. Jian T. et al. , Deep Learning for RF Fingerprinting: A Massive Experimental Study, 2020.
  3. Jan T. et al, Radio Frequency Fingerprinting on the Edge, 2021.
  4. Rehman, S. et al., Effect of Channel Impairments on Radiometric Fingerprinting, 2015.
  5. Elmaghbub, A. et al,, LoRa Device Fingerprinting in the Wild: Disclosing RF Data-Driven Fingerprint Sensitivity to Deployment Variability, 2021.

Author Response

Please see attached PDF with responses to both reviewers.

Reviewer 2 Report

The article addresses a consideration for radio frequency fingerprinting across multiple frequency channels.

General comment: The article is well-written and needs some improvement before acceptance.

1- The abstract should be rewritten again. There is not enough information about the methodology, proposed work, conclusion y comparison with other works in this part. Also, the abstract can be rewritten shorter but with more details and some numerical results.

2- The introduction was started with extra information about radio frequency and fingerprinting, where should be referred to some references. The background is very long and does not have details about recent works. The background should be extended to new published papers for recent years. In the background should be expressed the better state-of-art of new methods. The new references will also be examined in this part. I would like to see the articles for last and this year in this section.

3- Even though I understand that the results are scientifically sound, I recommend the author to do a major review on their text, which I believe can be heavily summarized in benefit of the readers.

4- I recommend to author to eliminate the extra parts from section 1 and merge the remaining of section 1 to section 2, where it is a better structure for a scientific article.

5- I don't find it costumery to have a review of the state of the art in a scientific paper. A brief overview, with the main references should be included in the introduction. I suggest that you merge sections 1 and 2. Furthermore, there is room to streamline you introduction.

6- I cannot see the details of the methods in this article. The methods were explained very short without preparing enough explanation and just with some figures. For example, sections 2.2 and 2.3 do not have any new information. They are completely well-know and can be referred to the references.

7- The block diagram in Figure 2 and 3 should be explained in a better way. There are many details, which have not been explained in the article. Also, if they are copy from some other articles or webpages, it should be referenced.

8- Simulation conditions are not well discussed. The approaches were illustrated only on some specific simulations, which is not enough to draw a complete and accurate conclusion about the method.

9- The evaluations in not enough. The proposed method should be compared with some well-known methods in this area.

10- Please, do not forget that the clarity and the good structure of an article are important factors in the review decision. Please read the paper carefully (again) and correct it in English.

Author Response

(The authors gave the same response as above.)

Round 2

Reviewer 2 Report

The authors addressed the comments and observations. The article can be accepted.